# REDUCING THE EFFECT OF INCOMPLETE ANNOTATIONS IN OBJECT DETECTION FOR HISTOPATHOLOGY

**Denys Kaliuzhnyi, Dmytro Fishman, Mikhail Papkov**
Institute of Computer Science, University of Tartu
Tartu, Estonia
`{denys.kaliuzhnyi,dmytro.fishman,mikhail.papkov}@ut.ee`

## ABSTRACT

Training neural networks for object detection usually requires decent amounts of data to produce great results. Apart from the image variety, the number of annotated objects is a crucial factor for success. In histopathology, the average annotation density is very high, resulting in resource-consuming data preparation for neural network training. We explore the effect of incomplete annotations in object detection. We show that modern object detectors, such as YOLO-v5, can effectively learn from histopathology datasets that lack up to 90% of annotations. Additionally, we suggest an easy model tuning setup to reduce the impact of incomplete annotations and enhance learning capability overall. We publish our code at `https://github.com/DenysKaliuzhnyi/yolov5`.

## 1 INTRODUCTION

Deep neural networks performing object detection tasks demand large quantities of images. Usually, each image is thoroughly annotated — every object has a label assigned by a human expert. The network is trained under the assumption that no object is missing. In histopathology, object density differs drastically from the popular natural image datasets, such as COCO (Lin et al., 2014), easily reaching several hundred per image. Thus, annotation efforts increase proportionally and can be costly and time-consuming.

This leads to a natural question: is it possible to label only a portion of the objects in each image without performance degradation from the model trained on 100% of data? A handful of research attempted to address this question. Zhang et al. (2020) proposed a background recalibration loss designed to downplay confidently misdetected objects. Wang et al. (2021) introduced a co-mining procedure with Siamese network generating pseudo-labels. Both methods were developed for RetinaNet (Lin et al., 2017) and reported more than 10% increase in $AP_{50}$ compared to the baseline when trained on 50% of annotations. However, they introduce additional hyperparameters to tune and require a substantial modification of the pipeline when transferring to another architecture.

In this work, we explore the influence of missing annotations on object detection performance in the histopathology domain. We hypothesise that modern neural networks, such as YOLO-v5 (Jocher et al., 2021), can be trained efficiently with a significant portion of missing annotations without sophisticated add-ons. Additionally, we find a set of hyperparameters that increases the overall model performance and reveal an instance that greatly aids in missing annotation settings.

## 2 METHODS

We used a small version of YOLO-v5 in all the experiments (we describe it in detail in Appendix A). To adopt the model for the histopathology domain and increase overall performance, we manually tuned hyperparameters. First, we considered vertical flip invariance and enabled the respective augmentation (`flipud`). Secondly, we tuned optimization parameters, such as learning rate, scheduler, and batch size. The exact values of the default and changed hyperparameters are listed in Table 2. Finally, to adjust the model for missing annotations, we decreased the objectness positive class weight (`obj_pw`) tenfold. It downweighted the impact of positive samples forcing the model to learn less from them. This change is crucial because we assume most false positives to be unannotated objects.

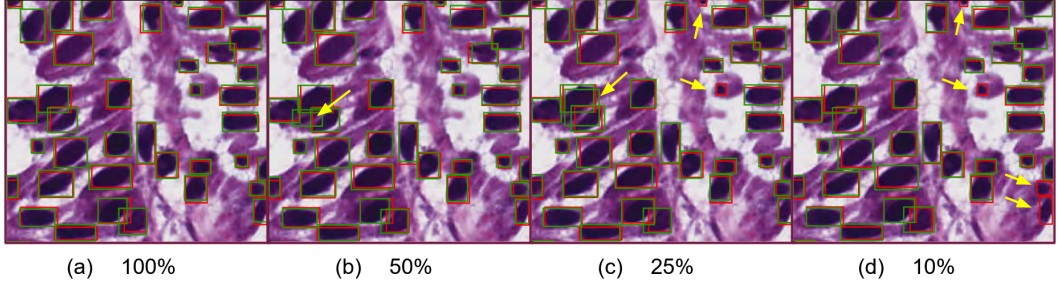

|     |      |     |     |     |     |     |     |
|-----|------|-----|-----|-----|-----|-----|-----|
| (a) | 100% | (b) | 50% | (c) | 25% | (d) | 10% |

Figure 1: Example test predictions of YOLO-v5 models trained on datasets with randomly sampled annotations. Subset sampling ratio is displayed below the image (100% is no sampling). Red boxes correspond to ground truth, green ones – to predictions. Yellow arrows highlight false predictions.

## 3 EXPERIMENTS

We trained all the models on MoNuSeg 2018 dataset (Kumar et al., 2017; 2019) — stained tissue images of various tumour-affected organs from multiple hospitals. The training set contains 30 huge-resolution whole slide images (WSI) with around $22\,000$ annotated cells. The test set consists of 14 images with $7\,000$ annotations. We follow Nguyen et al. (2021) for data preprocessing and extraction of $512 \times 512$ px training patches.

We uniformly sampled 50%, 25%, and 10% of annotations with five different random seeds for each image in the training set to simulate incompleteness. The number of images in the dataset remained the same. For each of the generated subsets as well as the complete set (further mentioned as 100% subset), we trained the baseline and tuned model. Then, the training results were averaged out for each percentage value across experiment seeds to yield more stable results (except for the 100% set).

We selected the model checkpoint with the best validation $AP_{50}$ and adjusted the confidence threshold to maximise $F_1$ score. Test evaluation results are summarised in Table 1. Annotation subsampling does not drastically affect the model's performance, even for the baseline. Additionally, the tuned model produced better results for all annotation subsets, including the complete one. Improved model reduced the gap between the 100% set and 10% subset from 3.3% to 1.9% $AP_{50}$ score. Example detection results are demonstrated in Figure 1.

YOLO-v5 appears to be robust to missing annotations assuming that the remaining labels are still adequate to learn object features (e.g., $2\,200$ objects in 10% set is still a decent volume). This robustness can be explained by a model's excessive box prediction before suppression. Additionally, an optimal confidence threshold is estimated on a complete validation set. Regarding model tuning, the key to the success of our method is downplaying the loss contribution of positive predictions.

Table 1: $AP_{50}$ detection score of models trained on data with missing annotation averaged over five runs with different random sampling seeds ($N\%$ in header denotes a sampling ratio per image).

|          | 100%             | 50%               | 25%               | 10%               |
|----------|------------------|-------------------|-------------------|-------------------|
| baseline | 0.903            | $0.892 \pm 0.003$ | $0.883 \pm 0.015$ | $0.870 \pm 0.015$ |
| tuned    | 0.909            | $0.907 \pm 0.002$ | $0.901 \pm 0.007$ | $0.890 \pm 0.012$ |
|          | (+0.6%)          | (+1.5%)           | (+1.8%)           | (+2%)             |

## 4 CONCLUSIONS

We demonstrated that YOLO-v5 is robust and can be further tuned for handling histopathology data with missing annotations. With high object density, it is possible to annotate just 10% of them while losing only 2% of $AP_{50}$ compared to the full annotation, and 50% can already yield near-equal results. This finding bears practical value, allowing pathologists to spend less time labelling data.

URM STATEMENT

The authors acknowledge that at least one key author of this work meets the URM criteria of ICLR 2023 Tiny Papers Track.

ACKNOWLEDGMENTS

This work was funded by Revvity, Inc. and the Estonian Ministry of Foreign Affairs Development Cooperation and Humanitarian Aid funds. The computational resources were provided by the High Performance Computing Center of the University of Tartu.

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

## A  TRAINING DETAILS

YOLO-v5 is a single-stage anchor-based object detector known to be very fast and accurate. Its backbone consists of CSP-DarkNet (Bochkovskiy et al., 2020), the neck is comprised of SPP He et al. (2015) layers and CSPNet (Wang et al., 2020), and the head is taken from YOLO-v3 (Redmon & Farhadi, 2018) model. YOLO-v5 provides a family of models that vary in size (network width and depth) with scaling analogous to the approach used in EfficientNet (Tan & Le, 2019). We took YOLO-v5s, a "small" family member that accounts for nearly 7.2M parameters and is generally considered to be very lightweight.

Table 2: Selected hyperparameters. Changed parameters highlighted in **bold**.

| Hyperparameter | Tuned value | Default value |
|---|---|---|
| lr0 | **0.005** | **0.01** |
| lrf | **0.05** | **0.1** |
| momentum | **0.977** | **0.937** |
| weight_decay | 0.0005 | 0.0005 |
| warmup_epochs | 3.0 | 3.0 |
| warmup_momentum | 0.8 | 0.8 |
| warmup_bias_lr | 0.1 | 0.1 |
| box | 0.05 | 0.05 |
| cls | 0.3 | 0.3 |
| cls_pw | 1.0 | 1.0 |
| obj | **1** | **0.7** |
| obj_pw | **0.1** | **1** |
| iou_t | 0.20 | 0.20 |
| anchor_t | 4.0 | 4.0 |
| fl_gamma | 0.0 | 0.0 |
| hsv_h | 0.015 | 0.015 |
| hsv_s | 0.7 | 0.7 |
| hsv_v | 0.4 | 0.4 |
| degrees | 0.0 | 0.0 |
| translate | 0.1 | 0.1 |
| scale | 0.9 | 0.9 |
| shear | 0.0 | 0.0 |
| perspective | 0.0 | 0.0 |
| flipud | **0.5** | **0.0** |
| fliplr | 0.5 | 0.5 |
| mosaic | 1.0 | 1.0 |
| mixup | 0.1 | 0.1 |
| copy_paste | 0.0 | 0.0 |
| weights | YOLO-v5s.pt | YOLO-v5s.pt |
| epochs | 300 | 300 |
| batch-size | **32** | **16** |
| imgsz | **512** | **640** |
| rect | disabled | disabled |
| noautoanchor | disabled | disabled |
| image-weights | **enabled** | **disabled** |
| multi-scale | disabled | disabled |
| optimizer | SGD | SGD |
| cos-lr | **enabled** | **disabled** |
| patience | 100 | 100 |

