# OpenReview forum: "Reducing the Effect of Incomplete Annotations in Object Detection for Histopathology"
_ICLR.cc/2023/TinyPapers — Submitted to Tiny Papers @ ICLR 2023_

### Official Review · Reviewer_wKd1 · 2023-04-01

**Confidence:** 3

**Summary Of Contributions:**

This work showed that a well-tuned YOLOv5 can learn effective to detect objects from histopathology datasets with limited annotations. The well-tuned model can perform substantially well even when only 10\% objects are annotated.

**Rating:**

Clear, Correct, and Reproducible (CCR): a submission which meets the reviewing criteria

**Strengths And Weaknesses:**

Strengths
- Limited training samples in object detection is an important problem, which manifests in many real-world application.
- The paper is generally well-written and is likely to be reproducible.
- The experimental results are encouraging.

Weaknesses
- It is unclear to me where the improvement came from: is it from the better hyper-parameter tuning, or from some inherent properties of YOLOv5?
- The authors are strongly encouraged to discuss more on the foundation, results, and extendibility of this work: can better hyper-parameters search translate to other architectures such as DETR and to other datasets; what are the principles to address the challenge of incomplete object annotation; why did the new set of hyper-parameter improved the results?


**Suggested Changes:**

See Weakness

---

### Official Review · Reviewer_DdQo · 2023-04-01

**Confidence:** 4

**Summary Of Contributions:**

This paper experimentally shows that incomplete annotation in object detection on histopathology datasets has marginal effects on the accuracy. Moreover, tuning the model can further decrease the performance gap.

**Rating:**

High Potential (HP): a submission which meets the reviewing criteria and has potential to make an impact on the field

**Strengths And Weaknesses:**

Strengths:
- The paper is well-written and easy to read.
- The authors provide sufficient details of the experiments and the method.
- The contribution of the paper seems very helpful for the domain experts since with around half the annotations, the results are almost unchanged.
- The suggested tuning set-up notably improves the performance on incomplete annotations.

Weaknesses:
- The experiments are done on only one dataset. It would be more insightful to see whether the argument holds for various datasets as well.

**Suggested Changes:**

I would suggest the authors investigate their argument also on segmentation tasks for their future works. Segmentation is another time-consuming task for experts.

---

### Official Review · Reviewer_UodZ · 2023-04-01

**Confidence:** 4

**Summary Of Contributions:**

This paper claims that neural networks can be trained on histopathology images with incomplete annotation for object detection without large sacrifice on accuracy. The hyperparameter configuration proposed minimizes the accuracy loss efficiently without modification on the architecture or the training pipeline.

**Rating:**

Clear, Correct, and Reproducible (CCR): a submission which meets the reviewing criteria

**Strengths And Weaknesses:**

Thank you for submitting this work to Tiny Papers @ ICLR 2023. The paper targets an important problem and presents an effective and efficient solution to deal with it.

Strength
- The paper describes the importance of the problem and the methodology clearly. The problem it focuses on is realistic and important, especially in the histopathology domain, where the lack of image annotations is even more substantial. It also includes and discusses previous relevant literature.
- The results of the experiment successfully prove their hypothesis that neural network can be trained efficiently in the absence of annotations without significant modifications.
- The paper follows the basic formatting requirements, page limit, as well as the ICLR code of conduct.

Weakness
- The model and hyperparameter settings are clearly described, but it would be better if the paper provides the code they used to reproduce the findings.
- The authors mention that the hyperparameters are tuned manually. However, in addition to the "tuned values" and "default values" listed in Table 2, more values of hyperparameters and the corresponding results are encouraged to be provided. Is the combination of hyperparameters specific to the model/dataset (i.e., can it be generalized to other models/datasets)? Is it possible to automate the tuning process to make the method more generalizable?

**Suggested Changes:**

The motivation to make modifications on hyperparameter configuration but not on the architecture or the training pipleline is interesting and practical. Here are some suggestions to further improve the paper.
- Apply the hyperparameter setting to larger/more diverse datasets to test its effecitveness
- Automate the hyperparameter searching process with grid search, random search, or other optimization algorithms with a larger hyperparameter space
- For future directions, try to annotate different portions of the images to see if it will make a difference on the accuracy

---

### Meta-Review · Area_Chair_H5fV · 2023-04-10

**Recommendation:** Invite to present
**Confidence:** 4

**Metareview:**

This paper shows that the limited annotations in object detection for histopathology has little effect on the detection accuracy using a fine-tuned YOLOv5. The problem is important and the results are encouraging. I believe this paper is interesting, clear, and correct, with some pros and cons listed below.

Pros:
1. The studied problem is meaningful and important -- the lack of image annotations is common in real-world applications such as the histopathology domain;
2. The reported results are promising and has the potential to be impactful in the histopathology domain;
3. The paper is well-written and easy to understand, and methodology details are sufficient.

Cons:
1. The authors are suggested to try some other datasets and see similar results can be found;
2. The authors are encouraged to study where the promising results came from, for example, they are from fine-tuned parameters or inherent property of YOLOv5, or some others;
3. The authors are encouraged to share the code for reproducibility.

**Summary:**

This tiny paper experimentally demonstrates that a fine-trained YOLOv5 can be effective to detect objects for histopathology image datasets with incomplete annotations. The problem is realistic and important, the experimental results are encouraging, and the paper is well-written.

**Reason For Not Giving A Higher Recommendation:**

1. The authors are suggested to try some other datasets and see similar results can be found;
2. The authors are encouraged to study where the promising results came from, for example, they are from fine-tuned parameters or inherent property of YOLOv5, or some others;
3. The authors are encouraged to share the code for reproducibility.

**Reason For Not Giving A Lower Recommendation:**

1. The studied problem is meaningful and important -- the lack of image annotations is common in real-world applications such as the histopathology domain;
2. The reported results are promising and has the potential to be impactful in the histopathology domain;
3. The paper is well-written and easy to understand, and methodology details are sufficient.

---

> ### Author Response · Authors · 2023-06-01
> **Consolidated feedback to reviews**
>
> Thanks to all of the reviewers for the comprehensive and valuable feedback!
> In this summary comment, we address the main questions/suggestions we received and mention changes made to the revised version of the paper.
>
> $\textbf{Q:}$ The authors are suggested to try other datasets and see if similar results can be found.
>
> $\textbf{A:}$ Initially, we tested the YOLOv5 model's performance in missing annotation settings using a single MoNuSeg 2018 dataset and obtained promising results. We are currently expanding our experimental settings to include other datasets. Unfortunately, the new results of the ongoing study are not yet polished enough to be included in the revised version of the paper. Nevertheless, we are willing to outline some of our validated insights in this comment to satisfy the readers' curiosity.
>
> We are currently testing the YOLOv5 detection model on the MoNuSAC 2020 dataset, which is known to be more complicated and diverse compared to MoNuSeg 2018. MoNuSAC's cell and tissue appearance diversity is higher, making detection on it more challenging. Moreover, MoNuSAC incorporates the classification challenge with an extreme class imbalance. Train data accounts for 31,000 nuclear annotations.
>
> Our experiments on the MoNuSAC 2020 dataset, analogous to those conducted on MoNuSeg 2018, show similar relative performance drops across different annotation completeness settings. MoNuSAC 2020 demonstrated the lower robustness, but the larger complexity of the task justifies that. The preliminary results are very promising, and we are working on further improving them.
>
>
> $\textbf{Q:}$ The authors are encouraged to study where the promising results came from, for example, they are from fine-tuned parameters or inherent property of YOLOv5, or some others;
>
> $\textbf{A:}$ We address this question in the revised version of the paper. In this comment we give a more elaborate answer.
>
> The inherent property of YOLOv5's robustness to missing annotations in the histopathology domain is evident from the baseline model results (i.e., YOLOv5 with no tuning). We explain that the model's robustness can come from dense predictions over the image grid. YOLOv5 inherently predicts an excessive volume of boxes and then suppresses the majority to keep the best instances. Then, during inference, a model can automatically select an optimal confidence threshold that best fits the validation set, which contains complete annotations.
>
> Regarding tuning, changes made to the majority of hyperparameters aimed to increase the overall performance, meaning across all the annotation completeness settings. Then, the main factor in reducing the effect of incomplete annotation is the objectness positive class weight factor (obj_pw) hyperparameter of YOLOv5, which we lowered 10 times. This hyperparameter effectively downplays the loss contribution of positive predictions, including false positives, most of which we assume to be missing annotations. The beneficial effect of this adjustment was observed in the validation curve (epoch vs MAP line plot). In a baseline scenario, the validation curve at some point starts overfitting the incomplete training set, which results in its positively skewed bell-shaped appearance. Hence, the best model checkpoint taken for evaluation resides in early epochs. With obj_pw being set to 0.1, the score drain is almost completely eliminated, and the validation curve tends to grow throughout the whole training period.
>
>
> $\textbf{Q:}$ The authors are encouraged to share the code for reproducibility.
>
> $\textbf{A:}$ We have added a link to the revised version of the paper for code reproducibility.
>
>
> $\textbf{Q:}$ Can better hyper-parameters search translate to other architectures?
>
> $\textbf{A:}$  It is generally hard to claim the ease of hyperparameter translation to other architectures. However, we can provide some suggestions in this comment, although we have yet to test them. Similar models, such as other members of the YOLO family (e.g., YOLOv7), may benefit from analogous tuning due to their similar architecture. For models more different from YOLOv5, we first suggest adjusting the objectness positive class weight factor or finding an alternative hyperparameter(s) that helps in mitigating the impact of false positive predictions.
>
>
> $\textbf{Q:}$ Automate the hyperparameter searching process with grid search, random search, or other optimization algorithms with a larger hyperparameter space.
>
> $\textbf{A:}$ It is important to note that most of the hyperparameters we tuned aimed to improve the overall model performance, so there are no special aspects that would differentiate our hyperparameter search from the one conducted under regular (complete) annotation settings. We emphasize that the objectness positive class weight factor (obj_pw) is the most crucial hyperparameter in mitigating the effect of incomplete annotations, and it should be tuned in priority using the either hyperparameter search method.

---

### Decision · Program_Chairs · 2023-04-10

Invite to present

---

> ### Author Response · Authors · 2023-06-01
> **Opt-in for archival**
>
> We would like to inform that we have uploaded the revised version of the paper and we wish to opt for the archival.